# Porous-Wall Titania Nanotube Array Layers: Preparation and Photocatalytic Response

**DOI:** 10.3390/nano13233000

**Published:** 2023-11-22

**Authors:** Dumitru Luca, Marius Dobromir, George Stoian, Adrian Ciobanu, Mihaela Luca

**Affiliations:** 1Faculty of Physics, Alexandru Ioan Cuza University of Iasi, 700506 Iasi, Romania; dumitru.luca@uaic.ro; 2Department of Exact and Natural Sciences, Institute of Interdisciplinary Research, Alexandru Ioan Cuza University of Iasi, 700506 Iasi, Romania; 3National Institute of Research and Development for Technical Physics, Dimitrie Mangeron Blvd., 47, 700050 Iasi, Romania; gstoian@phys-iasi.ro; 4Institute of Computer Science, Romanian Academy, Iași Branch, 700481 Iași, Romania; adrian.ciobanu@iit.academiaromana-is.ro (A.C.); mihaela.luca@iit.academiaromana-is.ro (M.L.)

**Keywords:** TiO_2_ nanotube array layers, pulsed anodization, nanotube wall porosity, dual-tree complex wavelet transform, 2D-Fast Fourier transform

## Abstract

Electrochemical anodization is already a well-established process, owing to its multiple benefits for creating high-grade titanium dioxide nanotubes with suitable characteristics and tunable shapes. Nevertheless, more research is necessary to fully comprehend the basic phenomena at the anode-electrolyte interface during anodization. In a recent paper, we proposed the use of sawtooth-shaped voltage pulses for Ti anodization, which controls the pivoting point of the balance between the two processes that compete to create nanotubes during a self-organization process: oxide etching and oxidation. Under these conditions, pulsed anodization clearly reveals the history of nanotube growth as recorded in the nanotube morphology. We show that by selecting the suitable electrolyte and electrical discharge settings, a nanoporous structure may be generated as a repeating pattern along the nanotube wall axis. We report the findings in terms of nanotube morphology, crystallinity, surface chemistry, photocatalytic activity, and surface hydrophilicity as they relate to the electrical parameters of electrochemical anodization. Aside from their fundamental relevance, our findings could lead to the development of a novel form of TiO_2_ nanotube array layer.

## 1. Introduction

Solar energy-driven photocatalysis has been attracting the interest of materials scientists for decades [1,2,3,4]. The Scopus database has documented up to 14,000 papers per year concerning titanium dioxide and 1400+ papers about titania nanotubes over the preceding 15 years. Low-dimensional nanostructured TiO_2_ materials, such as nanoparticles, nanowires, nanorods, nanobelts, nanotubes, and thin films, have been the subject of extensive research since 1972, following the early use of titania nanosized particles as low-cost, benign-nature pigments or UV light absorbers. The intense redox processes induced by UV/visible light irradiation of the titania surface of nanoparticles and 1D nanostructures have been used in environmental and energy-related applications such as photocatalytic degradation of organic pollutants in air and/or groundwater, the production of super-hydrophilic, self-cleaning, and antifogging surfaces, or the design of dye-sensitized solar cells, Li-ion batteries, super-capacitors [1,2], and optical waveguide spectroscopy for the study of biomolecule adsorption on surfaces [3].

Several modification techniques have been proposed to optimize materials’ optical and electrical characteristics and morphology. For instance, doping titania with anionic or cationic species or decorating the surface with noble metal nanoparticles have been introduced to extend the surface photoactivation range into the visible part of the light spectrum or to form heterojunctions, which allow an increase in the photoactivation span [4]. Especially one-dimensional self-organized titania nanostructures have attracted the interest of scientists and engineers because they retain the benefits of large specific area nanoparticles while introducing novel features such as easy recovery after use and peculiar electronic properties due to quantum size effects and band surface curvature [5]. These enhanced characteristics increase the surface interaction with its environment and provide new chemical reaction pathways.

The great majority of investigations on anodized titania nanotubes now use a potentiostatic configuration with constant anodization voltage. This type of anodization has been described as a method for producing “good quality” wall morphology [5], with a smooth wall surface and controlled physicochemical parameters. The synthesis of titania nanotubes via electrochemical anodization is linked to two main competing processes involving oxide formation (Equation (1)) and oxide dissolution (Equation (2)):

Ti + 2H_2_O → TiO_2_ + 4 H^+^ + 4 e^−^
(1)


TiO_2_ + 6 F^−^ → [TiF_6_]^2−^ (soluble) + O_2_
(2)


Additionally, the transport of the Ti^4+^ cations over the oxide layer results in an increase in [TiF_6_]^2−^ species by the so-called *ejection of transported cations* to the electrolyte [5]:

Ti^4+^ + 6F^−^ → [TiF_6_]^2−^
(3)


The important prerequisite for anodization to generate “good” titania nanotubes [1] is to select the suitable electrolyte and ensure the self-organization process at the metal/electrolyte interface by adjusting the balance between oxidation and dissolution.

A study by Chanmanee et al. [6] investigated the implications of replacing the constant voltage bias with pulsed anodization using a bipolar square-shaped voltage in experiments with electrolytes containing NH_4_F dissolved in distilled water or glycerol (C_3_H_8_O_3_, with dynamic viscosity—1.5 Pa·s). Well-defined morphologies with improved nanotube photo-response were identified by adjusting the voltage amplitudes between 25 V (for 180 s) and −4 V (for 30 s). During voltage reversal, the NH^4+^ absorption at the nanotube wall ensured periodic protection against oxide corrosion. Later, by varying the pulse parameters, a few morphologies with increased pollutant degradation ability have been identified [7]. Albu et al. [8] reported the formation of bamboo-like and interleaved nanolace-like structures after periodically switching the voltage between 40 and 120 V (duty cycle 0.28%) in their investigation of pulsed anodization with a solution of HF in ethylene glycol (C_2_H_6_O_2_, with dynamic viscosity—0.016 Pa·s) as an electrolyte.

The selection of an electrolyte is of the utmost importance since ionic mobility and electrical conductivity are inversely proportional to electrolyte viscosity [9]. As a result, modifying the electrolyte viscosity affects pore diameter, porosity, and the total production of titanium dioxide. Lower-viscosity organic electrolytes, such as those containing ethylene glycol (EG) as a solvent, allow for a faster dissolving rate during anodization, influencing the shape and crystallinity of nanotubes [10].

*Sawtooth-shaped pulsed anodization* (SSPA) was recently introduced in our prior works [4,11], with two notable advancements. First, it enables fine-tuning the threshold in the balance of the oxidation-dissolution processes during nanotube formation. Secondly, the relevant effects of voltage change over time are recorded as imprints along the nanotube length. Thirdly, by adjusting the chemical process rates and using appropriate electrolytes, nanotube array layers with customizable wall shapes and physicochemical properties can be formed.

Here, we extend our findings by showing how to prepare nanotubes with *wall porosity along their axis*, allowing for an increase in specific area. This was accomplished using the pulsed-voltage anodization option and the proper combination of electrolyte composition and anodization pulse shape. The possibility of tuning nanotube wall morphology, crystallinity, and ionic characteristics is investigated. Furthermore, these modifications are explored in terms of enhancements in photocatalytic response and surface hydrophilicity. The findings are expected to pave the way for a new class of nanotube array layers with applications extending not only to photocatalysis but also in related fields, such as porous waveguide-based sensing, which allows for the detection of a low amount of marker molecules in medicine and biotechnological applications, or for the monitoring of surface processes like analyte-binding molecules [3].

## 2. Materials and Methods

TiO_2_ nanotube array layers have been prepared in a two-electrode configuration anodization cell by ECA of 0.25 mm thick Ti foils (Sigma Aldrich, Darmstadt, Germany 99.7% purity), with an area of 2 cm × 1 cm. The Ti foils were polished using Nihonkenshi ESO P1000 grit emery paper, then cleaned for 15 min in an ultrasonic bath of acetone, ethyl alcohol, and de-ionized water before being air dried. The electrolyte was a 0.2 wt.% water and 0.15 M NH4F solution in ethylene glycol. The cathode was a 2 cm long, 2 mm diameter Pt wire placed 10 mm in front of the anode. Anodization was carried out at ambient temperature using magnetic stirring (400 rpm).

Biasing the anodization cell with periodic sawtooth-shaped voltage pulses (SSVP) was used to prepare the TNA samples. The Hameg HM8142 (Hameg Instruments, Mainhausen, Germany) dual-module power supply provided the bias voltage pulses, with the two-unit outputs connected in series. The first unit produced a constant voltage, *V*_1_, while the second one, operated in tracking mode, delivered a triangular output voltage with a repetition rate of 1/124 s^−1^. A low-power function generator (UDB1000 DDS, Uni-T, Chengdu, China) was used to drive the second unit. The duration of the falling edge of the voltage pulse was 100 ms.

The temporal variation of the bias voltage corresponding to one pulse is illustrated in Figure 1. Four samples were prepared for separate ECA tests utilizing periodic linearly increasing (sawtooth-shaped) voltages (Figure 1) as expressed by the equation:
*V*(*t*) = *V*_1_ + k*t*,
(4)

where *V*_1_ = 0, 10, 20, 30 V. The four sample categories were named TNA_0, TNA_10, TNA_20, and TNA_30 based on the value of the start voltage in each cycle, *V*_1_. Three distinct samples have been prepared in each category to test repeatability. The sample TNA_30, prepared using a constant voltage of *V* = 30 V, served as a reference for comparison with the pulsed anodization samples.

A two-channel acquisition system with a sampling rate of 0.1 s^−1^ was used for measuring the anodization current and voltage. The evolution of the anodization cell’s equivalent internal resistance, *R*(*t*), was calculated using *I*(*t*) graphs acquired by a 10-bit resolution data acquisition system [11].

To induce the TiO_2_ anatase structure, all the investigated samples were annealed in air at 400 °C for 1 h in a Barnstead F-1300 furnace (Barnstead Thermolyne, Dubuke, IA, USA) then cooled down with a 2.5 °C/min rate to room temperature. The structure of the investigated samples was inferred from the XRD data delivered by a Shimadzu LabX XRD-6000 diffractometer, Shimadzu, Kyoto, Japan) operated in the Bragg-Brentano configuration, using Cu Kα radiation (*λ* = 1.54059 Å) with 2*θ* varying from 20° to 60°. The layer morphology was investigated using a JEOL JSM 6390 FE-SEM instrument (JEOL, Tokyo, Japan) equipped with a Schottky electron gun cathode, operated at 80 kV accelerating voltage and 10 μA emission current.

The elemental and chemical content of the surface of the TNA layers were investigated using an Ulvac-Physical Electronics PHI 5000 VersaProbe XPS instrument (Ulvac-PHI, Chansen, MN, USA) with a monochromated Al Kα X-ray source (1486.6 eV) [12]. The photoelectrons were collected at a 45° takeoff angle. Following the normal procedure, the surface was analyzed using the Ti 2*p* and O 1*s* narrow spectra. Peak deconvolution was carried out using the MultiPak v.9.1 software [12]. Valence band XPS (VB-XPS) spectra were used to estimate the band gap of the samples and to discuss the degradation performances and the optical properties of the investigated nanotube layers.

The photocatalytic degradation rate of aqueous methylene blue (MB) solution at contact with the investigated samples was determined using spectrophotometric measurements (Shimadzu SPEC UV-Vis 2450, Shimadzu Kyoto, Japan), as reflected in the evolution of the absorbance peak at *λ* = 640 nm, in accordance with the ISO standard 10678:2010 [13]. To meet the Lambert-Beer law requirements, the starting concentration of MB in water was set to be 1 × 10^−4^ M. The TNA layers were first immersed in a 10 mL quartz cuvette filled with the MB solution for 12 h in complete darkness. Magnetic stirring was utilized to ensure that the adsorption/desorption processes were completely stabilized. Surface conversion was achieved by exposing the samples to a 9 W UV black-light lamp (Philips, Philips, Eindhoven, The Netherlands, *λ* = 365 nm, 1 mW/cm^2^ at sample surface measured by a PCE UV34 radiometer) for 12 h. During the bleaching procedure, the samples were kept under UV irradiation. The wavelength range of 400–750 nm was used to measure absorbance.

To evaluate the potential absorption effects of photolysis, independent tests of MB degradation were carried out in which identical settings for UV irradiation, time, solution temperature, and stirring were maintained in the absence of TNA samples in the same setup. Within the sensitivity limits of the spectrophotometer, no measurable photolysis effects were observed.

The surface wettability was assessed by measuring the contact angles (CA) of de-ionized water with TNA surfaces at RT (22 °C, 50% relative humidity) with a Data Physics OCA 15EC goniometer (Filderstadt, Germany). Water drops of 0.5 μL have been used to avoid gravity-induced shape alteration and to diminish the effects of water evaporation during the measurements. For surface photo-conversion, the samples were irradiated with a non-filtered high-pressure Hg lamp (*λ* = 253 nm, 120 mW/cm^2^).

## 3. Results and Discussion

The experimental *j*(*t*) curve shown in Figure 2 was obtained while preparing the reference sample (TNA_30) with a constant anodization voltage, *V* = 30 V. The shape of this *j*(*t*) plot is dictated by the nature of the electrolyte and the evolution of the oxide layer thickness. Both factors determine the intensity of ion fluxes (Ti^4+^, O^2−^, and F^−^) across the oxide layer [5,6]. Following the voltage switch-on, a rapid jump from 0 to 1.5 mA.cm^2^ (point A) occurs in the *j*(*t*) curve, followed by a nearly exponential decrease (the A-B interval), while a compact barrier oxide layer starts growing on the initially clean Ti metal surface. The current depends on the oxide thickness, *d*, according to the equation:
(5)j=a exp (bV/d)
where *V* is the voltage drop across the oxide layer and a and b are two experimental constants. At lower *d* values, the local electric field significantly accelerates ion transport, although this effect fades away over time due to oxide buildup. Instead of reaching a supposedly finite thickness (which would happen when the influence of field-related transport weakens up to thermal k*T* values), a change in this trend occurs (the BC interval in Figure 2) due to increased oxide dissolution. Because F^−^ ions migrate at twice the rate of (larger in size) O^2−^ ions, oxidation is dominated by oxide dissolution (Equation (3)), and layer thickness begins to decrease. The inter-pit areas then begin to build tree-like walls, eventually colliding and competing for the total discharge current [14]. This self-organization process is sustained until the available current is roughly equally distributed through all pore structures and self-ordered nanotube structures form under constant-voltage conditions. According to the ‘plastic flow’ hypothesis [5], further nanotube evolution is caused by oxide stress effects, which are at the root of the decrease in electrolyte resistance. The CD interval in Figure 2 should be represented as a saturation curve, but the slow decrease is due to the buildup of the soluble [TiF_6_]^2−^ product in the electrolyte during anodization [15].

The same processes are projected to recur in succession during SSPA, with significant modifications in current density evolution, which is now a function of both time and voltage. To disentangle the effects of the two parameters, we tracked instead the evolution of *R*(*t*) plots. Except for the first cycle, as shown in Figure 3, the *R*(*t*) plots feature two regions in each cycle: (i) an early stage, illustrated by a plateau of high electrical resistance, and (ii) a late stage, when conditions are met for the growth of uniform nanotubes. The higher resistance plateau is linked with oxide formation and ends at voltage values of about 23 V [11], due to the intensification of the effects of F^−^ ions penetration into the oxide layer. Next, the oxide growth is overcome by vigorous electrochemical dissolution, and a slow transition between these stages begins.

Large jumps in resistance are visible in stage (i) due to the randomness of the growth processes during the initial stages, related to the occurrence of tree-like nanowires on top of the inter-pit region at the beginning of nanotube formation [16], with special emphasis on samples with lower start voltage values (Figure 3a,b). The corrosion starts to produce detectable effects and stabilizes when *V* = 18.5 V. As previously observed [11], nanotube development is stable only when the voltage exceeds two-thirds of the end value (30 V), meaning that oxide formation lasts only approximately 24% (0–7 V) of each cycle, pore nucleation 50% (7–22 V), and tube growth 27% (22–30 V). Due to the identical linear increase in bias voltage, nanotube growth lasts 40% and 81% of the total cycle duration in samples with start voltages of 10 and 20 V, respectively. As a result, not only do unique morphologies emerge, as discussed further below, but a threefold increase in nanotube length from TNA_0 to TNA_30 should be noted (Table 1).

Furthermore, negative values were observed in the *j*(*t*) signal channel at the voltage pulse’s trailing edge (causing negative values of *R* in Figure 3a) upon switching the bias voltage between 30 V and 0 V in each pulse. This aspect is hardly observed for voltage drops of 20 V and 10 V, respectively. Current overshoots are caused by the release of charge carriers during voltage switching by a capacitor represented by the anode/electrolyte interface (with the oxide as a dielectric). A configuration like that depicted in Figure 3d might be used to characterize the internal circuit of the electrochemical cell in terms of circuit analysis. Here, *R_loss_* is the loss resistance of the interface, and *R_e_* is the resistance of the electrolyte.

In samples TNA_10 and TNA_20, the charge expelled by the capacitor upon voltage switching is smaller, due to both the smaller voltage drop and the lower loss resistance itself. Figure 3d displays a simplified equivalent circuit, in which *R_loss_* and *C* represent the resistance and capacitance of the oxide layer, respectively, and *R_e_* is the resistance introduced by the electrolyte solution. After fitting the simulated output to the experimental data using the National Instruments Multisim v. 14.3 simulation software, the resistance and capacitance values shown in Figure 3d were determined.

Figure 4 shows SEM images of the longitudinal views of the four TNA layer samples. The layers were fractured by scratching to reveal the axial morphology. Upper-layer exfoliation occurred in some cases because of weak binding of the top layers, caused by the thinner wall of the nanotubes at the open end.

The presence of well-defined nanotubes contradicts the findings of Zhang et al. [17], who claimed that a minimal concentration of around 0.4% NH_4_F is necessary for nanotube separation. One of the most striking findings of the present study is the presence of distinct pores along the length of the nanotube in all the layers prepared by pulsed anodization using EG as an electrolyte. All the samples feature a well-defined texture in a plane normal to the nanotube axis, as illustrated in Figure 4a,d,g,j. The cross-section SEM images reveal layered “nano lace-like” structures perpendicular to the nanotube axis because consecutive nanotubes contain adjacent pores in the nearly same spot. The number of sub-layers decreases from 5 to 6 in TNA_0, to 2 in TNA_10, and finally to 1 in TNA_20. Each sub-layer contains stacked short nanotubes featuring 3 to 4 nanopores each in sample TNA_0, but 10–28 nanopores in sample TNA_10 to 39 nanopores in sample TNA_20. Mechanical scratching causes the delamination of adjacent layers to be more visible in sample TNA_0 and gradually weaker in samples TNA_10 and TNA_20, where nanotube breakage occurs at random points in neighboring portions of the layers, as shown in Figure 4d,g. Some of the morphological features of the prepared samples are summarized in the first columns of Table 1, where the nanotube length for each sample is expressed as percents for the length of the reference sample.

The SEM images were processed in two separate steps to show the axial texture of the layers and to collect quantitative information on the nanotube axial texture. The dual-tree complex wavelet transform (DT-CWT) [18] technique was used to de-noise the SEM images by removing the spurious high-frequency components from the spectra of the input data. To make the original 1024 × 768-pixel SEM images stand out more, they were first improved in Corel Photo-Paint v. 12. To obtain the denoised filtered image equivalents, the DT-CWT of level 4 was applied to each image, followed by a 2D inverse DT-CWT. Finally, as shown in Figure 4b,e,h,m, a morphological closing handle was used to yield isolated black spots representing the location of pores in the input SEM images. Despite the lack of pores in sample TNA_30, which was synthesized under constant voltage settings, a small number of spots still appear in Figure 4k, but they are not linked to pore presence,but to the bamboo-like aspect [14] of smooth wall nanotubes in the sample TNA_30, prepared under constant-voltage conditions.

After calibration with the scale bars of SEM images, the 2D Fast Fourier Transform (2D-FFT) was used in the second stage to verify the texture along the tube axis and quantify the pore spacing. Figure 4c,f,i,l shows the resultant spectra, which show a slight reduction in spacings from 56 to 52 nm in samples TNA_0 to TNA_20 (Table 1). Furthermore, the DT-CWT transform has been applied to the full-size SEM images to double-check the spacing values. The previously noticed occasional spots in Figure 4k do not contribute notable spectral features along the Oy-axis, as seen in Figure 4l.

The total number of pores along the nanotubes is smaller than the total number of voltage pulses (60) in each anodization cycle. The differences in nanopore spacing and opening area in each sample are related to the magnitude of the delays of the onset of corrosion with respect to pulse start (Figure 3). If the oxidation rates are influenced by the local electric field intensity, variability between the three samples is most likely to occur, most notably during the first phases of oxidation in each cycle [11]. A full investigation of the variation in nanotube diameter along the full length was beyond the scope of this paper; however, the effect is most likely implicated in the emergence of the nanotube array arrangement since the nanotube diameter is solely controlled by the anodization voltage [19].

The XRD measurements revealed that the nanotube walls of the as-prepared samples have an amorphous structure. The value of annealing temperature, 400 °C, was chosen to develop as much anatase crystal structure as possible, knowing that the as-prepared samples are amorphous and above 500 °C, the rutile substitutes the anatase phase, featuring a much higher oxidative power. It is largely acknowledged that for nanoscale materials, when crystallite grains are larger than 30 nm, anatase becomes the most stable phase below 400 °C. For lower grains, the rutile phase develops predominantly even at lower temperatures [20].

The XRD plots depicted in Figure 5 show a dominating anatase TiO_2_ ordering. The intensity of the A(101), A(004), and A(105) peaks, located at 2*θ* = 25.3°, 37.9° ± 0.2°, and 53.8° ± 0.2°, respectively (according to JCPDS card no. 21-1272), increases in samples TNA_0 through TNA_30, denoting a gradual increase in nanotube crystallinity with the increase in nanotube size.

The TiO_2_ grain size was calculated according to the Scherrer formula [21] by using the A(101) signal in the XRD data as input:
(6)D=0.9λ/(β cosθ),
where β is the FWHM of the diffraction peak at the Bragg angle θ and λ= 1.54182 Å. The grain size of the porous samples, *γ*, increases gradually from TNA_0 to TNA_20 and remains a few percent larger than the reference sample TNA_30 (Table 1). The average grain size of nanotubes generated under the same electrical parameters in a glycerol-based electrolyte [4] is 40% smaller than in the present instance, a feature that is reflected in both the MB degradation rate and surface photoconversion, as discussed below. The titanium XRD signals, marked by star symbols on top of each plot in Figure 5, are always present due to the large penetration depth of the X-ray beam into the sample.

The weight percentage of the anatase phase, *W_A_*, was determined using the Spurr–Myers formula [22]:
*W_A_* = *I_A_*/(*I_A_* + 1.256 *I_R_*)
(7)

where *I_A_* and *I_R_* are the intensities of the anatase A(101) and rutile R(110) XRD peaks. The anatase/rutile concentration ratios in the porous samples listed in Table 1 are almost the same in porous samples (approx. 87%) and are 5% lower than in the non-porous sample.

The anatase A(004) peak partially overlaps with the intense titanium T(002) peak at 38.4° ± 0.2°. Furthermore, a weak signal of rutile phase R(100) at 2*θ* = 27.2° (JCPDS card no. 04-0551) is still visible and remains nearly constant in all samples. The increase in anatase phase concentration originates from the increase in the effective volume of nanotube walls, as demonstrated by SEM analysis [23].

The technique of X-ray photoelectron spectroscopy (XPS) was used to analyze the elemental composition and chemical state of atoms within a few atomic layers of the nanotube surface. The survey spectra, along with the high-resolution Ti 2*p* and O 1*s* XPS core-level spectra of all the samples, reveal no discernible variations in oxidation state or chemical composition. Therefore, Figure 6 and Figure 7 display only typical examples of such spectra. The survey spectrum in Figure 6 reveals no detectable nitrogen or fluorine atomic species within the sensitivity limits of our instrument. Figure 7 indicates the presence of the Ti^4+^ state exclusively with BE = 459.45 eV and 465.11 eV for the 2*p*_3/2_ and 2*p*_1/2_ spin-orbit components, respectively. The deconvolution of the O 1*s* peak reveals the presence of lattice oxygen in the Ti-O bond (BE = 531.02 eV) and another in the O-H group (BE = 532.5 eV).

The presence of the C signal in the survey spectra contrasts with the findings of Dronov et al. [24], who reported that, on the exterior of *smooth nanotube walls*, (i) carbon species are missing within the first atomic layers but rise with depth; (ii) the fluorine signal follows an opposite trend. Our survey spectra of *porous wall nanotube arrays* revealed a strong carbon signal originating in the first atomic layers. The elemental analysis discrepancies indicate the consequences of differing ion mobility during the preparation of anodized with smooth and porous nanotube walls, respectively. Additionally, the XPS spectra allowed us to derive the valence band maximum level of the layers. It is positioned at 3.25 eV below the Fermi level of the materials, regardless of sample morphology, as seen in the inset of Figure 6, indicating the presence of the highly oxidative anatase phase at the nanotube surface [4,25]. This contrasts with the rutile TiO_2_ phase, which has a band gap of 3.02 eV and is characterized by lower oxidative power.

The tests on the photocatalytic degradation of methylene blue solution were carried out in the same manner as described in more detail in our earlier research [8] and consisted of collecting UV-Vis light absorption spectra for each sample in subsequent steps. In the current investigation, similar shape absorption spectra were found, as for non-porous nanotube films [11], with a tiny absorption peak remaining after 180 min of reaction. The same conclusion holds true for the kinetic tendencies of MB bleaching. As in any first-order kinetic response, the plots of the logarithm of the normalized dye concentration vs. irradiation time show a linear variation in time (Figure 8). The bleaching kinetics of a sufficiently diluted dye solution (i.e., considerably smaller than 10^−3^ M) follow a dependence derived from the Langmuir-Hinshelwood law [26]:

ln *C*/*C*_0_ = k_r_K_e_
*t* = k′*t*,
(8)

where *C* is the dye concentration, k_r_ is the apparent reaction rate constant, and K_e_ is the apparent equilibrium constant for the adsorption of the dye on the catalyst surface.

After computing the rate constant, k′, using the absorbance spectra, the resulting values were corrected by taking the relative nanotube length into account (given in column 2 of Table 1). When the bleaching efficiency of the four samples is compared, the rate constants in reciprocal hours rise by a factor of 2.48 in sample TNA_20, of 3.5 in sample TNA_10, and of 3.00 in sample TNA_0, with correlation coefficients ranging from 0.989 to 0.998. The current results show that, when compared to the reference sample, the degradation efficiency of all nanoporous-wall samples increases by a factor ranging from 2.5 (in sample TNA_20) to 3.5 (in sample TNA_10).

The current results indicate that both the layer surface alteration and the incident light penetration depth play essential roles in the MB degradation process. Indeed, as Marien et al. [16] proposed, as a function of layer porosity, the light penetration depth should be multiplied by 1.5 to 2.0. If the number of charge carriers generated by UV photon absorption is proportional to the volume of absorbing material, the photocatalytic yield should rise monotonically with nanotube length. This explains, in our opinion, why the rate constant, k′, of the reference sample is considerably larger than that of the shorter nanotube layers.

An evaluation of the contact angle (CA) values before UV light exposure is important for addressing the impacts of changed morphology on layer wettability. The CA of the four samples ranges from 21° (for the sample TNA_10) to 29° (for the sample synthesized in steady-state conditions), as shown in Figure 9a. All the samples show a considerable decrease in CA after UV irradiation: the super-hydrophilicity threshold (5°) is reached after 8 min for all SSPA layers and two times slower for the reference sample TNA_30.

CA measurements were also utilized to trace the time evolution of surface wettability over the so-called *back-reaction time* when the layers were kept in darkness for 100 h and the surface gradually lost its photocatalytic ability. Prior to the back-reaction measurements, all the samples were exposed to UV light, as described in Section 2, until the initial CA values were significantly lower than 5°. The CA was then measured at 5-min intervals (during the first 15 min of the dark), followed by 25-min intervals thereafter (see Figure 9b). After 100 h of back-reaction, the sample TNA_0 reached the maximum saturation value of the contact angle, approximately 45°, and therefore lost the impact of UV irradiation completely. The sample TNA_20 had CA values close to 22°, indicating that full deactivation has not been reached. The final values of the CA of samples TNA_10 and TNA_20 after 100 h are very comparable to their pre-irradiation values.

A comparison of the results shown in Figure 9a,b shows that the saturation values of the CA of the samples TNA_0 and TNA_30 after the back-reaction are 15°–20° higher than before the start of photoactivation (Figure 9a), indicating a potential residual photoactivation induced prior to the start of the activation experiments. Moreover, despite having different mechanisms, the wettability and photocatalytic ability follow a similar tendency in the case of TiO_2_ materials [27]. This can be noticed when comparing the results of the apparently best-performing sample, TNA_10, which features the highest degradation rate and the most rapid surface conversion, but also the slowest evolution of the back-reaction.

## 4. Summary and Perspectives

We used periodic sawtooth voltage pulses for anodization and an electrolyte containing NH_4_F dissolved in a lower-viscosity organic electrolyte (EG) to prepare pristine TNA layers with longitudinal wall porosity. The crystallinity, elemental composition, and chemical state characteristics across all the layers were the same. The XRD results revealed a mixture of (dominant) TiO_2_ anatase with small amounts of rutile. The presence of the anatase phase was further confirmed by the VB-XPS data, which revealed a value of 3.25 eV for the band gap of the materials (specific for anatase TiO_2_) with no apparent variations across all samples. The presence of stoichiometric TiO_2_ composition was demonstrated by XPS analysis, with the Ti^4+^ oxidation state occurring solely in all samples and with non-observable fluorine species in the first atomic layers of the surface. A minor percentage of carbon was detected in the survey XPS spectra, originating in organic electrolyte.

The morphology of the TNA layers was demonstrated by monitoring the evolution of the internal electrical resistance of the discharge during anodization and the conditions to modify the balance of the oxidation-dissolution reactions in real time. An equivalent circuit for the non-stationary anodization was proposed, and the values of the electrical elements were calculated. Under these conditions, clearly defined axial textures demonstrated by the 2D-FFT spectra have been produced along nanotubes, as a function of the anodization voltage parameters, when using EG as an electrolyte. This resulted in improved photocatalytic performance of the textured samples and superior wettability characteristics compared to non-porous TNA layers. It was confirmed that wettability and photocatalytic ability follow similar trends upon UV irradiation; the layers with the highest degradation rate showed the most rapid surface conversion and the longest back-reaction. Such peculiar TNA layers may serve as scaffolding for embedded noble metal or organic nanoparticles, with the degradation rate greatly enhanced by surface plasmon resonance [28]. Porous nanotube arrays could be used in applications related to energy conversion and storage.

Because nanoporosity occurs at a spatial length scale far below the wavelength of the guided light, such optimized porous wall TNA layers look as promising candidates for photonic crystals in laser waveguides and demultiplexers [29] and as directional couplers for switching and splitting in Datacom. For such implementation, progress in ensuring pore distribution uniformity and density is required, which will be accomplished by experimenting with a broader range of electrolyte concentrations, voltage pulse amplitude, and frequency. A systematic, more quantitative evaluation of the surface areas of the studied nanotube layers is planned for the near future using the BET method; this will be an additional indicator of TNA layers’ efficiency in photocatalysis.

To develop an overall process model, more theoretical and simulation research is required.

## Figures and Tables

**Figure 1 nanomaterials-13-03000-f001:**
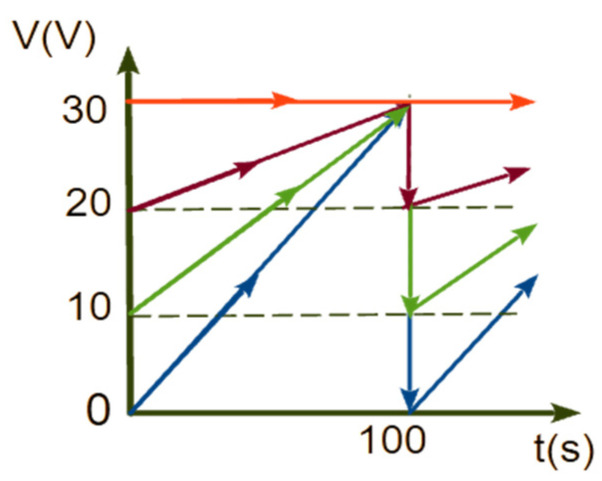
Temporal variation of the bias voltages for the anodization of samples: TNA_0 (blue), TNA_10 (green), TNA_20 (magenta), and TNA_30 (red).

**Figure 2 nanomaterials-13-03000-f002:**
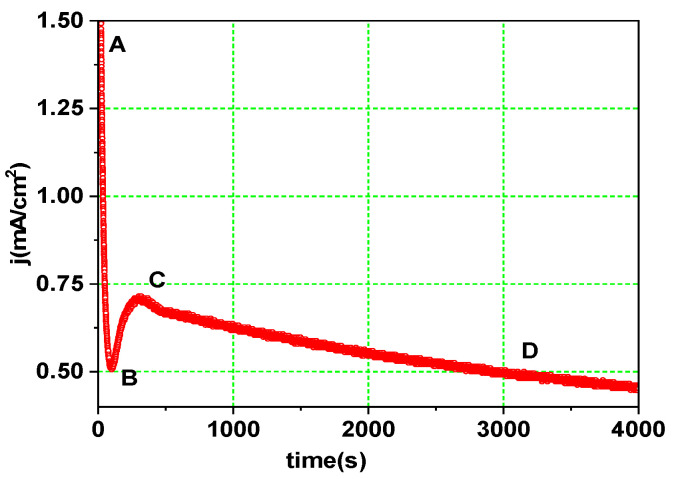
Temporal evolution of the anodization current density in the constant-voltage anodization. AB, BC and CD are intervals where oxidation, dissolution and steady-state nanotube growth prevail, respectively.

**Figure 3 nanomaterials-13-03000-f003:**
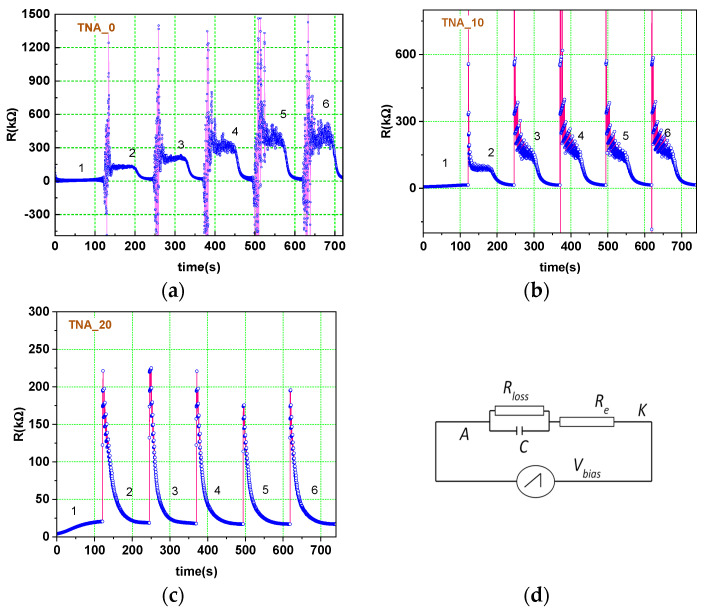
*R*(*t*) plots for the first six cycles of periodic SSPAwithin 0–30 V (**a**), 10–30 V (**b**), and 20–30 V (**c**) ranges; the equivalent circuit (**d**): *C* ≅ 50 nF, *R_loss_* ≅ 500 kΩ, *R_e_* ≅ 80 kΩ.

**Figure 4 nanomaterials-13-03000-f004:**
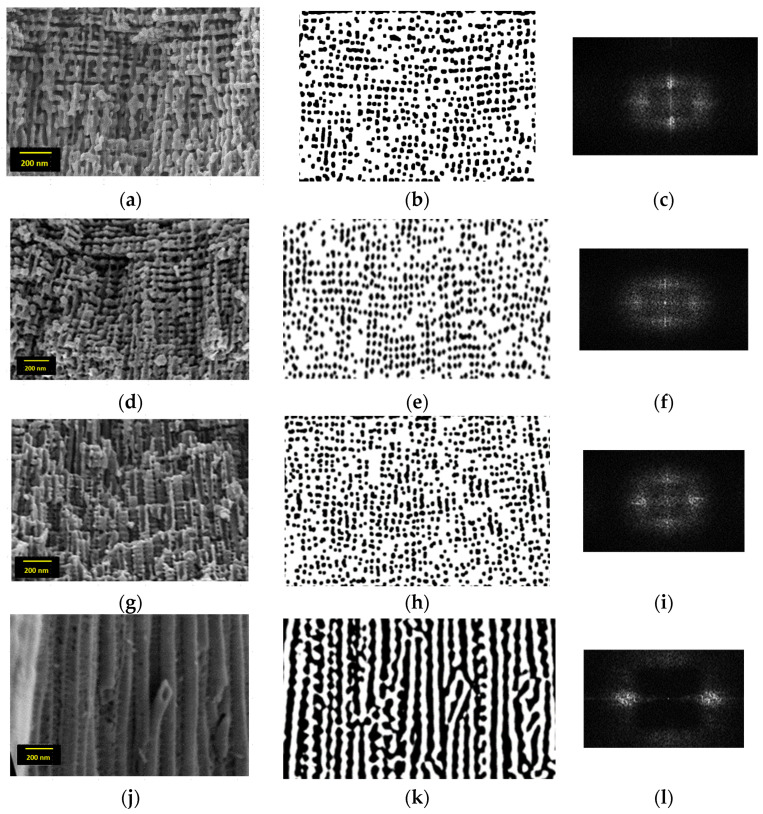
The SEM images of the investigated TNA layers (1st column (**a**,**d**,**g**,**j**)), the processed image correspondents (2nd column (**b**,**e**,**h**,**k**)), and the 2D—FFT spectra (3rd column (**c**,**f**,**i**,**l**)).

**Figure 5 nanomaterials-13-03000-f005:**
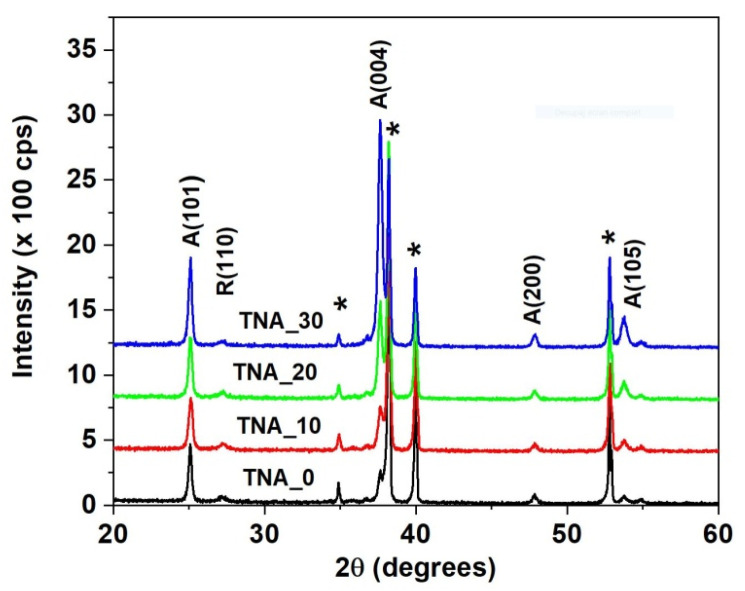
The XRD patterns of the investigated TNA layers. Titanium peaks are labeled with “*”.

**Figure 6 nanomaterials-13-03000-f006:**
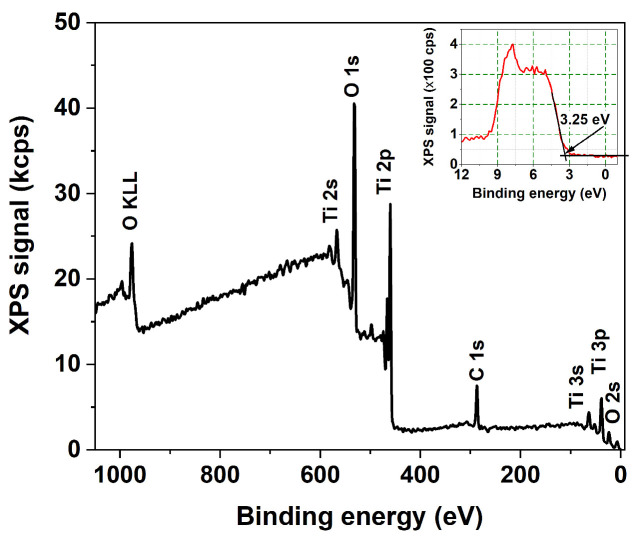
XPS survey spectrum with the VB-XPS spectrum; inset: a detail close to Fermi level.

**Figure 7 nanomaterials-13-03000-f007:**
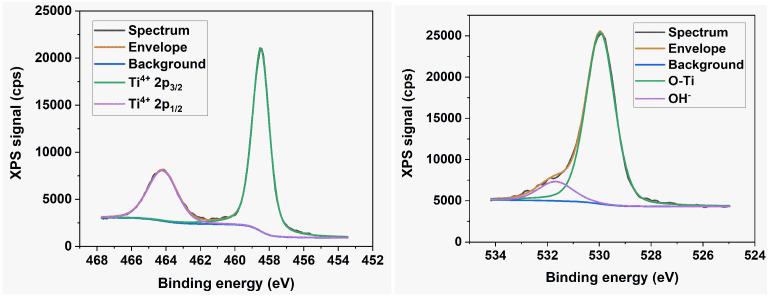
High-resolution Ti 2*p* core level and O 1*s* core level XPS spectra.

**Figure 8 nanomaterials-13-03000-f008:**
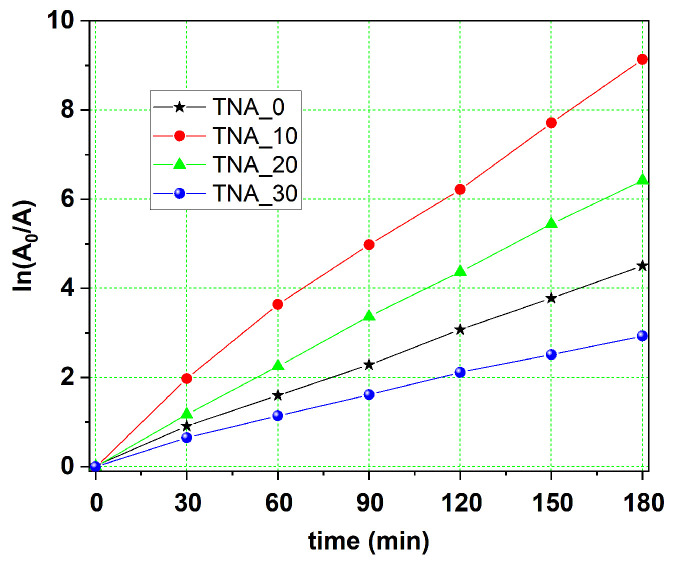
Semi-log plots of the intensity of the absorption peak vs. UV irradiation time.

**Figure 9 nanomaterials-13-03000-f009:**
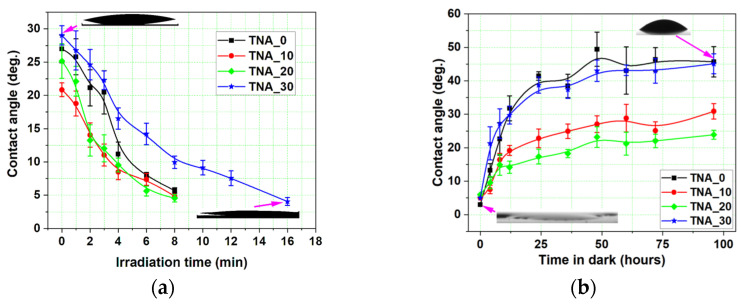
Temporal evolution of the contact angle: (**a**) during surface conversion and (**b**) during surface back-reaction. Arrows indicate the experimental points on the graph where the water drop images were taken.

**Table 1 nanomaterials-13-03000-t001:** The main morphological data and degradation rates of the investigated samples. *Rl*—nanotube relative length; <*N*>—average number of nanopores per nanotube; *s*—pore spacing; <*D*>—average nanotube diameter; WA—weight percentage of the anatase phase; *DR*—degradation rate; *R*^2^—correlation coefficient; *CDR*—corrected degradation rate.

Sample	*Rl*	<*N*>	*s* (nm)	<*D*> (nm)	*γ* (nm)	*W_A_*	*DR* (h^−1^)	*R* ^2^	*CDR* (h^−1^)
TNA_0	0.17	25	56	39	33.6	0.87	0.87	0.999	5.11
TNA_10	0.25	37	55	39	34.0	0.85	1.49	0.997	5.96
TNA_20	0.36	58	52	40	34.9	0.88	1.52	0.999	4.22
TNA_30	1.00	-	-	41	30.2	0.92	1.70	0.994	1.70

## Data Availability

The data presented in this study are available on request from the corresponding author.

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
