# Peer review of "Porous-Wall Titania Nanotube Array Layers: Preparation and Photocatalytic Response"

_nanomaterials, 2023, doi:10.3390/nano13233000_

Round 1

Reviewer 1 Report

Comments and Suggestions for Authors

The authors have described the fabrication of titania nanotubes for photocatalytic applications, this work may be interesting, but why was titania chosen as the material for photocatalytic, and what is its advantage to develop a nanotube?

The first part of the result section is an experimental section because it describes how the nanotubes are growing. For example, what is the use of introducing Figure 4?

SEM images of Figure 5 show poor resolution and scale bars are not visible. Please describe these images in more detail.

The XRD and XPS results are shown but the proper scientific discussion is still missing.  The authors can improve and combine Figure 7 to 2x2 figures. What is the advantage of introducing XPS measurements?

The authors should also show the CA images.

Comments on the Quality of English Language

Please write this more fluently 

Reviewer 2 Report

Comments and Suggestions for Authors

In this paper, the authors show here that by selecting the appropriate electrolyte and electrical discharge parameters, a nanoporous structure may be produced as a repeated pattern along the nanotube wall axis. By varying the voltage pulse slope and start values, conditions for adjacent nanotube separation were also investigated. We deal with the findings in terms of the morphology, crystallinity, surface chemistry, photocatalytic activity, and surface hydrophilicity of the nanotube array layers as they correlate with the electrical properties. Aside from the obvious fundamental implications, our findings could lead to the development of a new type of TiO2 nanotubes. The idea behind this is interesting. However, I still have quite a number of concerns in this manuscript. There are times where there are not enough data to support the conclusions of the author. Please see some of the major concerns below.

1.The information for figure 5 - the SEM images of the investigated TNA layers is not enough. The authors should give much more information about this. So the readers can get its reproducibility, for example what is the SEM resolution in each figure and etc.

2.  The authors should give much more information about the novelty of this paper, especially the effect of implements and selecting the appropriate electrolyte and electrical discharge parameters, which applications can be used this device?

3. The fabrication tolerance analysis, which can offer a good guide for the fabrication requirement, and the key parameters, need to be added in the results section.

4. To enhance the comprehensiveness of this work, a comparison table highlighting key differences between this study and other state-of-the-art research must be included

5. More references need to be included in the introduction part to understand the applications of optimized optical waveguides.

a.     Photonic Crystal Flip-Flops: Recent Developments in All Optical Memory Components- Materials, 2023

b.     Optimizations of Double Titanium Nitride Thermo-Optic Phase-Shifter Heaters Using SOI Technology

- Sensors, 2023

6.  Much more discussion about the results should be given in this paper, especially the author needs to provide enough physicals mechanism analysis about the results.

 Author Response

Reviewer 3 Report

Comments and Suggestions for Authors

The work "Porous-wall titania nanotube array layers: fabrication and photocatalytic response" is undoubtedly of interest, but in its current version, it is not well written methodologically.
1) In particular, the introduction mentions that "We extend this research by demonstrating how to develop a previously unknown form of nanotube porosity, allowing for a considerable increase in specific area via the pulsed-voltage anodization option and a proper combination of electrolyte and pulse parameters," but as a reviewer, it was not clear to me what exactly constitutes the previously unknown form of nanotube porosity. The novelty of the work should be clearly described.

2) The paper discusses the multicomponent synthesis of samples, replete with technical and technological details. Since the synthesis is very complex and multi-stage, I would suggest the authors consider the possibility of providing a clear flow chart or graphical description of the synthesis, which would better help readers understand the stages of the synthesis method.

3) There's no mention of surface area measurements (BET analysis) or detailed porosity characterization, which are critical for understanding the photocatalytic performance. Is it necessary?

4) It is not entirely clear how successfully the synthesis can be repeated.

5) Only one dye (methylene blue) is used for the photocatalytic degradation tests, which does not provide a broad enough assessment of photocatalytic activity. Is the use of just one dye sufficient for drawing general conclusions?

6) It's not entirely clear what the shortcomings and technological challenges are with the presented method.

7) Also, please describe the nomenclature of the samples (TNA_0, TNA_10, etc.) more clearly.

Overall, the work requires a major revision, but if the presentation of the material is improved, it could attract much greater reader interest.

Comments on the Quality of English Language

English is fine for me but I'm not a native speaker 

Round 2

Reviewer 1 Report

Comments and Suggestions for Authors

The authors have addressed on comments. This manuscript may be suitable for the publication.

Comments on the Quality of English Language

Some typos are observed.

Reviewer 2 Report

Comments and Suggestions for Authors

My notes have not yet fully addressed the revisions in the manuscript.

  1. What are the tolerances of the fabrication results?

  2. Why were the recommended references not included in this paper? These references are crucial for showcasing an optimization method that has not been introduced in the paper at all.

Reviewer 3 Report

Comments and Suggestions for Authors

The work has significantly improved. Perhaps I would discuss the prospects of BET analysis findings for future research in conclusion, in other aspects the revision is completely satisfactory to me.

Round 3

Reviewer 2 Report

Comments and Suggestions for Authors

The new version can be published.